# Connections between Attitudes towards Muslims, Meta-Prejudices and Religion-Related Factors among Finnish Christian Background Youth

Kanerva Lattu

Department of Practical Theology, University of Helsinki, 00100 Helsinki, Finland; kanerva.lattu@helsinki.fi

**Abstract:** Finland, traditionally characterised as a Christian country, is becoming increasingly more culturally, religiously, and ethnically diverse, which invites investigation on intergroup relations. This study examines the connections between attitudes towards Muslims, meta-prejudice—the ingroup's expectations of the outgroup's negative evaluations of the ingroup—and religion-related factors among Finnish Christian-background youth (N = 140). I analysed the survey data, gathered in 2019 and 2020, by scrutinising the distributions, descriptive statistics, and statistical inferences (Spearman's RHO, *p*-value). Most participants (73%) hold a positive attitude towards Muslims. There were no statistical differences between groups of different religiosities. Prejudiced Christian background youths were more likely to expect Muslims to evaluate Christians negatively. However, fairly over a half of the participants thought that Muslims evaluate Christians negatively. Cross-faith friendships and knowing religiously others through one's parents are connected to a pro-Muslim attitude. The findings are discussed from a Finnish societal standpoint and from the social psychology perspective of reducing prejudice.

**Keywords:** Finland; adolescent; attitude; prejudice; meta-prejudice; religion; Muslims; Christians

## 1. Introduction

Today, about a third (35%) of Finns are acquainted with a Muslim (Pew Research Center 2018, p. 22), but this will likely change. According to the Pew Research Center, Finland's Muslim population will be five-fold by 2050 (Pew Research Center 2015, p. 50). Currently, about 100,000 to 120,000 Muslims live in Finland (Pauha and Konttori 2021, p. 238), predominantly in major cities and the capital area (Martikainen 2020). Growing migration increases cross-group contact, giving rise to stereotypes—assumptions about characteristics that apply to most of the members of a social group—and prejudices/negative attitudes—negative evaluations of a target group (Stangor 2000, pp. 1–2, 6; Brown 2010). Stereotypes usually contain an evaluative connotation; characterisation is either positive or negative (Brigham 1971). According to scholarly and empirical research, prejudices can lead to bad treatment towards the target group members (see, e.g., Allport 1935; Allport 1954, p. 51; Welply 2018; for a review, see Brown 2000, pp. 281–85). Indeed, anti-Islam attitudes, Islamophobia and hostility against Muslims among the majority have been documented by scholars in Finland and elsewhere in Western Europe (see., e.g., Welply 2018; Bleich 2011; Pauha and Ketola 2015; Ketola 2011, p. 71).

Relatively little research exists on how the majorities expect the minorities, such as Muslims, to evaluate them. Expected negative evaluations might reduce willingness to be in contact with outgroup members (Finchilescu 2005). For future positive intergroup relations, it is important to gain knowledge about the factors that might influence relations between groups. The youth are at the frontline of ensuring functional cross-group processes in the future of an increasingly diverse society. The present article quantitatively examines attitudes towards Muslims and their connection to meta-prejudice—the ingroup's assessment of the outgroup's negative evaluation of the ingroup—and to religion-related

factors among Finnish adolescents whose religious background corresponds to the Finnish majority's: (Lutheran) Christianity. The research questions are:

1.　What kind of differences can be found in the data between participants' attitudes towards Muslims?
2.　What kind of differences can be found in the data between participants' expectations of how Muslims evaluate Christians?
3.　What kind of connections can be found in the data between attitudes towards Muslims and meta-prejudice?
4.　What kind of connections can be found in the data between attitudes towards Muslims and religion-related factors?

## 2. Research Context

Despite growing diversity, a notable number of Finns, 3.9 million out of 5.5 million, are members of Christian communities. Roughly 69% of Finns are members of the Finnish majority church, the Evangelical Lutheran Church of Finland (SELCF 2021). The membership statistics do not reveal the diversity of Finnish Christians as the numbers include multiple different worldviews. Many members do not actively practice their religion (e.g., Ketola 2020; Salomäki 2020), and younger people in particular do not place much importance on church membership or the teachings of the Church (Salomäki 2020). Currently, 1.5 million Finns are not affiliated with any registered religious community (Sohlberg and Ketola 2020, p. 52). Some of them are likely to be adherents of Christianity or non-Christian religions and are not registered in any established religious organisation.

Nevertheless, scholars argue that Lutheran ethos is present in the Finnish culture: Lutheran Christianity strongly characterizes Finland (see, e.g., Sinnemäki et al. 2019; Helkama and Portman 2019). For example, the Finnish Evangelical Lutheran Church has a long history of organizing activities for children and youth, and it is an essential collaborator of the established Finnish educational system. This is demonstrated, for instance, when a parish worker visits a school to set up a morning assembly. In addition, Lutheran confirmation school is popular among the young; in 2019, 50,100 adolescents (77% of all Finnish 15-years-olds) participated in Lutheran Confirmation education, and 44,800 of them were confirmed (Hytönen 2020, p. 207).

Lutheranism's influence can be seen in the conceptions of Finnish Muslim youth. Young Muslims associated Finnishness with Christianity. (Pauha 2018). Also, some Finnish youth with a Christian background merge Muslimness with nationality or ethnicity. The Muslim stereotype held by some of the Christian background youth reveals perceived status differences or power inequalities between groups when they think, for example, that Muslims have low status. (Lattu and Innanen 2022, in review). Studies by the Pew Research Center show that 62% of Finns think that Islam and the Finnish national culture and its values are incompatible (Pew Research Center 2018, p. 66). Thus, the research context is Finnish society wherein Lutheran Christianity is rooted, and Islam is seen as foreign to Finnishness.

## 3. Prejudices, Religion and Meta-Prejudices: General Research and Research in a Finnish Context

There is an extensive amount of research on prejudices, stereotypes, and attitudes between majority groups (usually dominant) and minorities (usually dominated), both internationally (for a review, see, e.g., Stangor 2000) and at the Finnish national level. Previous empirical studies show that there is a connection between religiosity and prejudices; more religious people tend to evaluate outgroup members negatively (see, e.g., Rowatt et al. 2014, pp. 171–72; Batson et al. 1993, pp. 310–11, 329). Some scholars track the connection between prejudices and religiosity to group processes, such as religious group identification (Rowatt et al. 2014, p. 184; Hall et al. 2010), but various dimensions of religiosity, such as religious identity, might not be salient in all situations to result in evaluations of outgroups (Chaves 2010). Burch-Brown and Baker (2016) argue that the conclusion that religiosity

and prejudice interrelate is too overgeneralized, and future studies should adopt more sophisticated methods that note the diversity of religious communities, cultural context, and social learning.

Early scholarly research suggests that long-lasting positive intergroup relations reduce prejudices (Allport 1954, pp. 488–91) and increase positive intergroup attitudes. A meta-analysis conducted by Davies and others shows a significant association between cross-group friendships and positive intergroup attitudes. (Davies et al. 2011). Furthermore, Pettigrew and Tropp suggest that positive cross-group contact reduces prejudices towards a target group and promotes tolerance and willingness to encounter other outgroups (Pettigrew and Tropp 2000).

It was not until recently that researchers positioned meta-stereotypes and meta-prejudices as an essential part of intergroup relations. Social psychologists have adopted the concept of the 'meta-stereotype' from a work by Vorauer et al. (1998). Vorauer and her colleagues define meta-stereotypes as the ingroup's appraisals of how the (relevant) outgroup sees the ingroup. Meta-stereotypes, like stereotypes, consist of psychological qualities or traits and their evaluative connotation (i.e., positive or negative evaluation). Researchers use the term 'meta-prejudice' to refer to an expected negative evaluation (see, e.g., Putra 2016; Gordijn 2002).

Vorauer and others found, for example, that White Canadians believe they are being evaluated negatively by Aboriginal Canadians. White Canadians thought that the Aboriginal Canadians perceive White Canadians as e.g., 'selfish', 'spoiled', and 'racist'. (Vorauer et al. 1998). Sigelman and Tuch studied how Black Americans thought they are being evaluated by White Americans. They thought that they are viewed as violent and lazy. (Sigelman and Tuch 1997). Kamans et al. (2009) found that youth with Moroccan background, i.e., dominated group, thought they are seen as 'criminal', 'aggressive', 'extreme Muslim', and 'loitering teenager' in the eyes of the indigenous Dutch, i.e., dominant group. (Kamans et al. 2009). Indeed, scholars are interested in studying meta-stereotypes with groups that have a different status, i.e., minority and majority, dominant and dominated groups.

Finchilescu (2005) reviewed previous research on the connections between prejudices and meta-stereotypes and suggested that these connections are ambiguous. According to her review, it seems that the majority's awareness of the minority's struggle might lead to the majority expecting to be negatively evaluated. People producing negative stereotypes might anticipate negative evaluations in return. (Finchilescu 2005). For example, Vorauer et al. (1998) showed that prejudiced White Canadians thought that they were evaluated negatively by Aboriginal Canadians, but also unprejudiced participants held a negative meta-stereotype because they were aware of how the majority had discriminated against them (Vorauer et al. 1998). Kamans et al. (2009) proposed that by assimilating the negative meta-stereotype, an individual may begin to behave accordingly.

There does not exist, to the best of my knowledge, research that examines means to reduce meta-prejudice.

According to Finnish scholars, Finns have negative attitudes towards Islam (Pauha and Ketola 2015, p. 100; Ketola 2010, pp. 47–48). In Finnish studies, a modest connection between anti-Islam prejudices and religiosity has been found. Commitment to Christian dogma, nationalistic attitudes, leaning towards right-wing politics, and rejecting the value of equality were connected to anti-Islam attitudes. On the other hand, no relationship existed between religious activities and negative attitudes towards Islam. (Pauha and Ketola 2015, p. 103). In Ketola's study, religiosity was only modestly connected to anti-Islam prejudices (Ketola 2011, p.73). However, the younger generation (born between 1990 and 1999) has less anti-Islam attitudes than the preceding generations (Ketola 2020, p. 28). A study by Koirikivi et al. (2021) showed that Finnish adolescents did not seem to be prejudiced towards immigrants or other specific target groups, but the young were prejudiced against those whose lifestyle, from their point of view, differed from their own.

Lattu (2021) and Lattu and Innanen (2022, in review) studied meta-stereotypes in a Finnish context. The meta-stereotype held by Finnish Muslim background youth reflects their position as a minority: Muslim adolescents think that they are evaluated negatively, as a norm-breaking group by the Finnish dominant group, Christians. I followed the guidelines set by previous studies on (meta)stereotypes/prejudices: I selected the groups by considering the status differences in Finnish society. Youth whose background is in (Lutheran) Christianity represents the majority, whereas Muslims are a relatively new minority group.

## 4. Method

### 4.1. Research Instrument

This study is part of research that focuses more broadly on intergroup relations in a Finnish context. I present the components of a research instrument relevant to this study and describe how it relates to research questions.

1.  The participants were asked what their affiliation was to the Christian denomination (unidentified, baptized, attended confirmation preparation, or neither). As the research studies the opinions of Finnish youth with Christian backgrounds, it is important to know their affiliation.

2.  In this study, a negative stereotype—negative appraisal of a group—is associated with a negative attitude towards a target group or with being prejudiced (e.g., Brown 2010; Stangor 2000, p. 1; Allport 1954; for a different view, see Verkuyten et al. 2019). Stereotypes were measured using open-ended questions in which participants could write three characteristics of their own choice. The English translation of the introduction of the response to the question measuring stereotypes was: 'I think most Muslims are ... ' The answers were quantified according to the evaluative connotation of the characteristics. The attitude towards Muslims is equated with assigning a positive or negative stereotype to Muslims. This procedure is described in the analysis design.

3.  Following Pauha and Ketola (2015) and Ketola (2011), religion-related factors are (a) a categorical variable that represents self-reported individual differences in believing in God and (b) continuous variables that represent self-reported involvement in religious practices (see also Saroglou 2014, p. 5). I combined believing in God (or not) and involvement in religious activities because they reflect different aspects of religiousness. Moreover, religion-related factors refer to (c) continuous variables that represent involvement in cross-faith relationships. To test the association between religion-related factors and attitudes towards Muslims, (a) the respondent's self-assessment of belief in God was measured using a checkbox question with six response options. Four groups were identified: believers (in God), believers in a higher power (but not in God), agnostics, and atheists. Also, (b) involvement in religiously motivated practices and, reflecting the idea of contact hypothesis (Allport 1954), involvement in cross-faith relationships were measured by forced-choice Likert scale questions with options 1–5.

4.  A meta-stereotype is the ingroup's appraisal—psychological qualities or traits—of how the outgroup sees the ingroup. Meta-prejudice refers to an expected negative evaluation. Meta-stereotypes were measured using an open-ended question and subsequently quantified in the same way as a stereotype measurement. The English translation of the response to the introduction of the question measuring meta-stereotypes, based on studies by Vorauer et al. (1998) and Kamans et al. (2009), was: 'Before people get to know each other, they often have an assumption of each other. What do you think Muslims think of Christians before they get to know them? I think that Muslims think we Christians are ... ' In the introduction of the questionnaire, the participants were reminded that they represent people whose religious background is in Christianity, whether their membership is important to them or not and regardless of their opinion on God.

### 4.2. Data

In the established Finnish education system pupils have the right to participate in specific religious studies in accordance with their religious traditions. In 2019, there were 13 different curricula for different religious traditions at the basic education level. A child or adolescent has the right to be educated in classes of a minority religion if they are a member of that religious community or if their cultural heritage corresponds to that religion. Children and adolescents can study secular ethics if they do not belong to a registered religious organization or have no religion class of their religion in their municipality's school. Everyone, regardless of their religious tradition, is allowed to participate in the majority's religion class, which de facto is Lutheran studies (see e.g., Ubani et al. 2019, pp. 7–11; Sakaranaho 2014).

This study utilised data gathered in 2019 and 2020 in upper comprehensive and secondary schools. I contacted several teachers—who teach Lutheran classes—around Finland via email. The adolescents whose teachers (N = 5) were willing to assist the researcher participated in this research. The teachers forwarded the link to the electronic questionnaire to the class attendees, who filled in the questionnaire during class. All class attendees were asked to fill in the questionnaire, but those who had not been confirmed or baptized were told to skip the question measuring meta-stereotypes. This way, it was possible to identify those whose religious background most likely corresponds to the majority's: (Lutheran) Christianity. Before collecting the data, I obtained permission from municipal authorities, headteachers, and guardians.

Through this procedure, 220 respondents from southwest Finland (N = 128), southeastern Finland (N = 31), the capital area (N = 8), and eastern Finland (N = 53) returned the questionnaire. The respondents were mostly upper comprehensive school ninth graders, aged around 14 to 16. The respondents from eastern Finland were from upper secondary education, aged about 16 to 17.

### 4.3. The Design of the Analysis

Those whose religious background most likely does not correspond to Christianity (N = 14) were excluded from the analysis. Not all the respondents answered the questions; 12 did not perform the task measuring meta-stereotypes, and seven skipped the question measuring stereotypes. I excluded respondents' answers from the analysis if they ignored the instructions, used swear words or gave other answers that clearly indicated not taking the task seriously. I also excluded responses if they did not reflect the a priori definition of a meta-stereotype or stereotype, such as statements that (Muslims are) welcomed in Finland. I excluded the data of 39 participants because respondents obviously did not take the task seriously or provided attributes that did not reflect the a priori definitions of a meta-stereotype and stereotype.

Next, I considered the evaluative connotations of the characteristics. Negative characteristics addressed unlawful behaviour or behaviour that breaches human rights (e.g., terrorist, criminal, violent), a nonchalant attitude towards others (e.g., rude, impolite) or provided words that explicitly suggest that something is unfavourable (e.g., bad believer). Positive characteristics described sociability: the ability to get along with others, respect for others and taking others into account (e.g., kind, respectful), or traced similarities between people and human value (e.g., valued, ordinary, like us). The latter consideration is also based on the notion that young Muslims in Finland think Christians evaluate Muslims as breaking norms (Lattu 2021); therefore, an inclusive characterisation by Christian background informants is considered affirmative.

The connotation was unclear in some cases. To elaborate on the evaluative connotation of such characteristics, I conducted a task in an upper secondary school among 14 adolescents. The task was to choose which characteristics in the data selected by the researcher were negative, positive, and which were neutral. The task showed the existence of characteristics that were either negative, positive, or neutral, depending on the evaluator. I labelled the characteristics as uncertain. The characteristics that the adolescents judged as



negative or positive were categorised in the same way in this study. Table 1 presents all the Muslim stereotype characteristics and the meta-stereotype characteristics that the Christian adolescents provided. The characteristics are categorised according to their evaluative connotations.

**Table 1.** Negative, positive, and uncertain meta-stereotype and Muslim stereotype characteristics provided by Christian background youth categorised according to their evaluative connotations.

| NEGATIVE characteristics provided as a meta-stereotype | POSITIVE characteristics provided as a meta-stereotype | UNCERTAIN characteristics provided as a meta-stereotype |
|---|---|---|
| Racist, prejudiced, judgmental, ill-judged, contemptuous, adversarial, stupid, incredulous, silly, selfish, hypocritical, egoistic, self-centred, Islamophobic, consider themselves better, believe in wrong God, not real believers, heretic, sinner, unholy, bad believer, heathen, religious infidels, weird, bizarre, odd, rude, impolite, unemphatic, dangerous, unapproachable, shameless, antisocial, hostile, privileged, more valued, superior, invader, white trash, annoying, wrong kind, critical, alcoholic. | Friendly, helpful, honest, nice, kind, good, great, easy-going, fun, fair, well-intentioned, warm, amicable, sympathetic, trustworthy/loyal, co-operative, easy-going, open-minded, approbative, happy enthusiastic, glad, perky, normal ordinary, like them, like everyone else, like others, basic, equivalent, equal, valued, lovely, well-behaved. | Religious, believers, quiet, introverted, reserved, calm, rich, white, not believers, does not practice religion, irreligious, laid-back with religion, prayer, preacher, free, liberal, talkative, different, special. |
| **POSITIVE characteristics provided as a stereotype** | **NEGATIVE characteristics provided as a stereotype** | **UNCERTAIN characteristics provided as a stereotype** |
| Nice, polite, kind, fun, friendly, social, respectful, well-mannered, polite, thoughtful, fair, helpful<br>Well-intentioned, good, hospitable, compassionate, trustworthy, easy-going, gentle, ordinary, normal, basic, like everyone else, like us, like others, equivalent, as valued as us/others, valued, unique, open-minded, approbative, tolerant, reasonable, unprejudiced, happy, enthusiastic, glad, interesting, intriguing, competent, undangerous, beautiful, rational. | Too deep in their religion, violent radicalised, terrorist, belongs to ISIS, criminal, killer, bombing, scary, mean, threatening, uncomfortable, malevolent, legitimises bad behaviour with religion, unapproachable, weird, bizarre, peculiar, odd, poor, out of work, outsiders, not well behaved, selfish, discriminated against, rides camels, misunderstood. | Extremely religious, religious, believers, different, emotional, quiet, traditional, dark skin, not Finns, from Africa/Asia/Middle East/East, women wear scarfs, immigrants, foreigners, speaks Arabic. |

Next, I quantified the data according to Vogt et al. (2014). To reach an interval scale, I coded the answers on a scale of 1 to 5 in which the participant produced:

1. Negative characteristics;
2. Negative and uncertain characteristics;
3. Uncertain characteristics;
4. Positive and uncertain characteristics;
5. Positive characteristics.

For example, if a participant provided three characteristics that described sociability, the answer was coded 5. If a participant wrote three characteristics, two of which addressed sociability and one religiosity, the code for this answer was 4. To obtain a clear interval scale of 1 to 5, participants who wrote both positive and negative characteristics were excluded from the analysis (N = 26). Vogt and others included "do not know" and "unsure" as "neutrals" in the centre of the scale (Vogt et al. 2014, pp. 29–30).

I quantified stereotype and meta-stereotype expressions separately. This procedure had the following continuous variables:

(1) v3: asses the differences between participants' expectations of how Muslims evaluate Christians (negative meta-stereotype—positive meta-stereotype).
(2) v4: asses the differences between participants' attitudes towards Muslims (negative stereotype—positive stereotype).

v4 was also used as an independent categorical variable with two groups (prejudiced/negative attitudes—unprejudiced/positive attitudes) and v3 as a dependent continuous variable.

Finally, I quantified the stereotypes provided by 140 informants, and 110 also provided meta-stereotypes. I merged the quantified data with the quantitative data, that is, with the answers of the items saved in the SPSS software (Version 28). I analysed the merged data using SPSS software (Version 28). All variables were not normally distributed and therefore I used tests that were not sensitive to nonlinearity. I conducted the non-parametric Kruskal–Wallis test to compare the differences between attitudes towards Muslims among the participants who differ in their believing in God. I examined the connection between attitudes towards Muslims and involvement in religious practices and in cross-faith relationships using correlation analysis and Spearman's Correlation Coefficient. I used Mann–Whitney's U-test to compare the differences between prejudiced and unprejudiced youth's anticipations of Muslims' evaluations of Christians.

## 5. Results

### 5.1. Attitudes towards Muslims and Expectations of How Muslims Evaluate Christians

The first research question was: What kind of differences can be found in the data between participants' attitudes towards Muslims? To address the question, the distribution of variable v4, 'attitudes towards Muslims'—was scrutinized. The distribution showed that most of the informants, 73% (n = 103), appraised only positive (n = 67) or positive and uncertain (n = 36) characterisations of Muslims. Only 19% (n = 26) evaluated Muslims negatively. The attitudes of a little less than a tenth (8%, n = 11) of the youths remained uncertain.

The second research question was: What kind of differences can be found in the data between participants' expectations of how Muslims evaluate Christians? The distribution of variable v3—'expectations of Muslims' evaluations of Christians'—revealed that those Christian background youth who also provided their meta-stereotypes (N = 110) presumed that Muslims view Christians negatively; 65% (n = 71) of them thought that Muslims evaluated Christians completely negatively (n = 47) or negatively, but the characterisations included uncertain characteristics (n = 24). Approximately a third (32% n = 35) expected Muslims to evaluate Christians positively. Only 3% (n = 4) provided uncertain characteristics. There was a moderate, statistically significant correlation between variables 3 and 4. Spearman's RHO is 0.305 ($p$ = 0.001). A more negative attitude is connected to expected negative evaluations.

As for the third research question—what kind of connections can be found in the data between attitudes towards Muslims and meta-prejudice—the participants, who also provided their meta-stereotypes (N = 110), were divided into two groups according to the evaluative connotations of their expressed characteristics. Group one (N = 23) provided negative characteristics (scale options 1 and 2). This group consisted of youths who were prejudiced against Muslims. The second group (N = 79) expressed positive characteristics (options 4 and 5). This group included unprejudiced participants. Independent continuous variable, v3—'expectations of Muslims' evaluation of Christians'– ranged in values from 1 (highly negative) to 5 (highly positive). Mann–Whitney's U-test (U = 1237.500, $p$ = 0.005) showed that the differences between the groups were statistically significant.

I tested the medians of the groups ($\chi^2$ = 7.241, $p$ = 0.007, df = 1. The median of prejudiced youths was 1 whereas the median of the unprejudiced youth was 2. The prejudiced youth are more likely to expect negative evaluations from Muslims.

### 5.2. Attitudes towards Muslims and Religion-Related Factors

The fourth research question was: What kind of connection can be found in the data between attitudes towards Muslims and religion-related factors? I conducted a Kruskal–Wallis test to compare the differences between attitudes towards Muslims among Christian background youth who differ in their belief in God. The participants were clustered into

four categories: 1. believers (N = 38), 2. believers in a higher power (N = 20), 3. agnostics (N = 37), and 4. atheists (N = 35). The differences between the groups were not statistically significant ($\chi^2$ = 1.948, $p$ = 0.583, df = 3). An individual relationship with or opinion of God was not connected to attitudes towards Muslims. I also tested the medians of the groups ($\chi^2$ = 0.582, df = 3, $p$ = 0.901). The differences between the groups were not statistically significant.

I examined the connection between attitudes towards Muslims, involvement in religious practices and in cross-faith relationships using correlation analysis and Spearman's Correlation Coefficient. Next, Table 2 shows descriptive statistics of involvement in religious practices and cross-faith relationships, correlations between them, and attitudes towards Muslims (range 1–5).

**Table 2.** Descriptive statistics of involvement in religious practices and cross-faith relationships, correlations between them, and attitudes towards Muslims (range 1–5) among Christian-background adolescents. (Spearman's RHO, 2-tailed).

| | M (N) | SD | Range | RHO with Attitudes towards Muslims | Sig. |
|---|---|---|---|---|---|
| Attitudes towards Muslims | 3.9 (140) | 1.4 | 1–5 | - | - |
| I have friends whose faith is different to mine. | 3.5 (138) | 1.3 | 1–5 | 0.190 | 0.025 |
| My parents have friends or acquaintances whose faith is different to that of our family. | 3.2 (140) | 1.2 | 1–5 | 0.174 | 0.040 |
| I personally know people who have different faiths. | 3.3 (138) | 1.4 | 1–5 | 0.104 | 0.225 |
| My inner circle consists of people who belong to a different religion to mine. | 2.7 (138) | 1.5 | 1–5 | 0.096 | 0.263 |
| I pray. | 1.8 (139) | 0.99 | 1–5 | 0.063 | 0.453 |
| I personally ponder religious matters. | 2.0 (139) | 1.0 | 1–5 | 0.106 | 0.216 |
| I read the Bible. | 1.5 (138) | 0.85 | 1–5 | 0.132 | 0.123 |
| I participate to mass more often than just during Christmas. | 1.6 (139) | 0.85 | 1–5 | 0.066 | 0.437 |

The results show that hardly any correlation existed between attitudes towards Muslims and involvement in religious practices. Religiousness does not seem to be connected to the attitudes towards Muslims among the participants. Cross-group friendships with people of different faiths correlated weakly with attitudes towards Muslims (RHO = 0.190, $p$ = 0.025). Also, having cross-faith contact through the participant's parents had a weak correlation with attitudes toward Muslims (RHO 0.174, $p$ = 0.040). On the contrary, there was hardly any correlation between variables "I personally know people who have different faiths" and "My inner circle consists of people who belong to a different religion to mine" and attitudes towards Muslims.

## 6. Discussion

This article presents the results of a quantitative analysis that examined the connections between attitudes towards Muslims, meta-prejudice, and religion-related factors among Finnish Christian-background adolescents (N = 140). First, the differences between participants' attitudes towards Muslims and the differences between participants' expectations of how Muslims evaluate Christians were elaborated. The data of this research benefitted from survey data collected in 2019 and 2020. The research context was Finland, a diverse

society, but traditionally characterised as a Christian (Lutheran) country, in which scholars have perceived a contradiction between the Finnish majority and minority Muslims (Pew Research Center 2018; Pauha and Ketola 2015; Ketola 2010). In this study, altogether 73% of the participants described Muslims using positive characteristics. Only 19% evaluated Muslims negatively. The result of the present study supports earlier findings: the younger Finnish cohort (born 1990–1999) showed relatively less anti-Islam attitudes than older generations (Ketola 2020).

The participants who were prejudiced against Muslims expected Muslims to evaluate Christians negatively in comparison to the unprejudiced participants. The differences between the groups were statistically significant. This suggests that prejudiced people are more likely to expect others to evaluate their own social group negatively (for a discussion on this, see Finchilescu 2005). This was only one part of the case in which over half of the participants thought that Muslims see Christians in a negative light. It is relatively straightforward that the anti-Islam attitudes perceived in Finland were reflected in negative meta-stereotypes held by Muslim adolescents in Finland (Lattu 2021), but why does the majority expect negative evaluations from the minority? Are anti-Islam attitudes also reflected in the negative meta-stereotypes held by Finnish Christian-background youth: if the young are aware of the anti-Islam attitudes of others and bad treatment against the subordinate group, do they expect Muslims to evaluate Christians, ergo the majority, negatively? To answer this intriguing question, adopting qualitative methods may enable us to understand why Finnish Christian-background adolescents think that Muslims evaluate Christians negatively. However, this study sheds light on the relationship between prejudices and meta-prejudices.

Cross-group friendships with people of different faiths and having cross-faith contact through one's parents were weakly connected to positive attitudes towards Muslims. However, there was hardly any correlation between attitudes towards Muslims and generally knowing religiously others. The *quality* of knowing has previously been associated with reduced prejudices: a meta-analysis conducted by Davies and others showed that cross-group friendships relate to lower intergroup prejudices (Davies et al. 2011). According to the contact hypothesis, the contact should be positive in nature (Allport 1954)—friendships are presumably a positive thing. Possibly, contact through parents is also positive in nature.

This study reflects the results of previous studies: religious practices do not play a major role in the life of Finnish adolescents (see Salomäki 2020). The attitudes towards Muslims of the participants who differed in their relationship to God did not vary. There were hardly any correlations between involvement in religious practices and attitudes towards Muslims. Koirikivi et al. (2021) suggest an individual-centred approach to investigating the prejudices of the young. In their study, they showed that the prejudices of the young rise from the fear of conflict with the ones that they think hold a lifestyle that differs from their own (Koirikivi et al. 2021).

It is also possible that the components of religiousness were not salient enough in this situation to influence evaluations; Chaves argued that situations where different dimensions of religiosity, such as religious identity, are not sufficiently salient or relevant to influence evaluations of other groups could exist (Chaves 2010). In general, people tend to think that their relationship with God is their own business and does not concern other people (see, e.g., Batson et al. 1993, p. 25). According to Burch-Brown and Baker (2016), it could be considered whether the connection between prejudice and religiosity is more context-dependant than innately being religious.

In this context, it is possible that the result could be interpreted through a perspective that acknowledges the status difference between Christians—the majority—and Muslims—the minority. This is because scholars have perceived a contradiction between the Finnish majority and Muslims (Pew Research Center 2018; Pauha and Ketola 2015; Ketola 2010; Lattu 2021), and the Muslim perceptions (characterisations) held by Finnish Christian youth reflect perceived power inequalities between Christians and Muslims (Lattu and Innanen 2022, in review).

This research has several strengths, including the topical research question, multi-faceted study methodology and findings worth noticing, yet I am aware of the following significant limitations. Due to the correlational nature of this research, the study does not imply that cross-faith friendships or having cross-faith relationships through one's parents lead to positive attitudes towards Muslims. Indeed, some other factors could explain the correlations between the variables and attitudes towards Muslims. Nevertheless, previous studies have shown that cross-group friendships are associated with tolerance (Davies et al. 2011), so it is plausible that in this study, cross-group friendships may also be independently connected to pro-Islam attitudes.

Furthermore, the non-probability sample is not representative of demographics and does not reflect the population of Christian adolescents in Finland. A relatively recent study has shown that a little over a third (35%) of Finns have personal contact with Muslims (Pew Research Center 2018). Most Muslims in Finland live in the capital area or inland (Martikainen 2020). As the participants of this study were mostly from Western or Eastern parts of Finland, they might not be familiar with Muslims. However, people tend to evaluate others they have never met (Allport [1954] 2000, p. 28). Furthermore, the teachers who helped the researcher in the data gathering process might have been especially interested in the study's topic, which might have been reflected in their teaching—perhaps the research sample is biased in this respect. Although the sample was large enough to enable statistical tests, generalisation of the results of this study must be done with considerable caution. Given that the current quantitative research is based on a study that was predominantly qualitative in nature, the sample size is small.

It is also important to consider why a notable number of participants did not provide their answers to the open-ended questions according to the task. Possibly, the task, especially related to meta-stereotypes, was difficult for some participants. Most likely, some of them did not believe the assignment to be serious or important. Perhaps others inferred that because they do not know a Muslim, they should not respond. Presumably, some participants did not want to express stereotypical thinking. As some informants pointed out in their feedback for the researcher, they did not want to categorize people. It is likely that some of them did not wish to provide their negative thoughts. There are many likely reasons why some informants ignored the instructions, and thus it is reasonable to presume that not only those who responded hold positive attitudes towards Muslims.

Moreover, I did not ask how committed to Christianity the participants were. Since previous studies have shown that the association between prejudices and religiosity is connected to group processes (Rowatt et al. 2014; Hall et al. 2010), it would be useful to adopt methods to measure several different dimensions of religiosity, such as group identification. This limitation should be considered when the results of this study are interpreted, but it does not affect the validity of this research for the following reason. This study suggests that religiousness is not connected to prejudices, which is ipso facto an interesting result, although I acknowledge that a self-reported religious commitment would shed more light on the connection between religiosity and prejudice. Nevertheless, (meta-)stereotypes can be studied without knowing participants' commitment. Their beliefs reveal information on how the members of a particular group, whether their group membership is essential or not, see the outgroup or believe to be seen in the eyes of the outgroup. Even when they disagree with it, group members with low identification may nonetheless be aware that others could label them as social group members (see Ellemers et al. 1997; Branscombe and Ellemers 1998).

My restricted study does not contribute to offering empirical evidence to explain the connections or demonstrate how (meta-)prejudice impacts behaviour, but it is nevertheless relevant: it contributes to the continuing discussion regarding the inclusion (or exclusion) of Muslims in the West. Furthermore, this study adds to the earlier findings of Finnish research. Younger generations do not appear to have anti-Islam attitudes. Furthermore, given that young Muslims in Finland believe they are being judged harshly by Christians, ergo the majority, the replacement of beliefs with knowledge is timely. This study opens new paths

for research hereafter. For example, further studies are needed to understand the connection between positive evaluations and expected negative evaluations. Hopefully, my limited, but topical study invites fellow researchers to scrutinize this relatively indeterminate field.

**Funding:** This research was funded by the OLVI Foundation and Kirkon tutkimuskeskus–the Church Research Institution, Finland. The APC was funded by University of Helsinki.

**Institutional Review Board Statement:** Ethical review and approval number were waived for this this study due to the Finnish legislation, or The Finnish National Board on Research Integrity (TENK) does not require an ethical review if the study does not entail the processing of personal data. As a result, I did not organise Ethical Review.

**Informed Consent Statement:** All informants were informed that their anonymity is assured, why the research is being conducted and how the data is used. In Finland, written consent from informants is not required if the research does not include the collection of personal data. Because the data collection took place in a school setting, guardians were informed about the study and written consent was obtained from them.

**Data Availability Statement:** The research data should not be shared because the researcher informed the guardians of the study participants that the data was only accessible to her.

**Acknowledgments:** I wish to thank my supervisors and the Research seminar participants for their helpful comments on prior versions of this paper. Open access funding provided by University of Helsinki.

**Conflicts of Interest:** The author declares no conflict of interest.

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
