# Peer review of "Connections between Attitudes towards Muslims, Meta-Prejudices and Religion-Related Factors among Finnish Christian Background Youth"

_religions, doi:10.3390/rel13111027_

Round 1
Reviewer 1 Report
-This is an interesting study of a topical subject. Introduction / literature review section is informative and well referenced. Rationale for research and research task are properly argued for.
-Key concepts are mostly well defined and consistently used. However, sometimes “attitudes” seems to be used as a synonym to prejudices/evaluations, and RQ uses this concept as well. Please define attitude & its relation to other concepts. The readability of the text and conceptual clarity could be improved by avoiding synonyms.
-Research task is relatively complex - it would help the reader to see research questions that elucidate the research task already in the end of Introduction, after the description of research task. I advice to divide the research task into several research questions, and to report the results in the results section under sections titled according to the research questions
-The order of writing in section 3 could be better. I would suggest beginning by research on metasterotypes and metaprejudices, and minority-majority aspects, then moving on to empirical research on religion & metaprejudices and finishing with research from Finland
-Section 4: “I did not want to label the majority as Finns because 150 Muslimness is not incompatible with being a Finn.” – I think it is self evident that the Christian majority group in this study could not be labeled as Finns and this sentence should be deleted
-The research instrument “forces” participants to express stereotypes. The questions are formulated In a way that it is impossible to answer to them in a way that would not express a stereotype. This should be critically discussed as parts of the limitations of study, both as a research ethical question and as question of validity. Maybe this could also be one of the reasons some of the students did not want to answer to these questions according to the instructions and were excluded from analysis.
-I don’t see the relevance of explaining Finnish RE system here – one or two sentences related to the RE system is enough to explain the context where data was collected
-How was informed consent of students acquired, and did they have the opportunity to withdraw/not participate in the research?
-“The quantitative analysis was built on the quali- 237 tative categorisation of the characteristics given by the participants who filled in the open- 238 ended questions of the questionnaire according to the task (N=167).” ß please explain in another way, difficult to understand what this means
-In a article targeted for international audience please avoid expressions as “Eastern upper secondary school” – I suppose this refers to eastern parts of Finland?
-Results are based on basic level analysis (descriptive statistics and group comparisons). Data and methods as well as results are a little hard to follow and conventions of reporting statistical analyses are not fully followed. The median figures look like SPSS outputs. Quite much space is used in the results section to reporting the medians. However, there are some interesting results.
A big part of limitations focuses on the limitations of the study. These are important to discuss and some more could be added, related not only to the possible biases and underrepresentativeness of data but the research instruments and analysis. However, more condense writing style should be developed so that there would be space to discuss the implications of the results as well (for education, policy, etc).
-The article is written in relatively fluent English but throughout the manuscript there are grammatical errors, sentences that are a little clumsy and confusing, and misspelled words (cross/gross) – language revision is recommended.
Reviewer 2 Report
Dear author(s)
I enjoyed reading your paper. Particularly, I appreciate the way that you cleaned up the data and how you organized it for analysis. I'm happy to see that research on meta prejudice is taking place in Finland. This is still a relatively new area of research in general. I commend you for your work.
I only have one suggestion for you. Go through your references and ensure that they are consistent with APA.
Otherwise, your paper was great! Wish you the best with your research.
